# Preparation and Macro-Micro Properties of SBS/REOB Modified-Rejuvenated Asphalt

**DOI:** 10.3390/polym14235071

**Published:** 2022-11-22

**Authors:** Jin Li, Li Zhu, Xiaozhou Yan, Chongsheng Xin, Miaozhang Yu, Degang Cheng

**Affiliations:** 1School of Transportation Civil Engineering, Shandong Jiaotong University, Jinan 250357, China; 2Engineering Department of Jinan Kingyue Highway Engineering Company Limited, Jinan 250013, China

**Keywords:** recycled engine oil bottom, SBS modifier, rheological properties, the modification-rejuvenation mechanism

## Abstract

To solve the problem that waste oil residues cannot be utilized and to reuse the aged asphalt, suitable modifiers were selected to compound the aged asphalt with waste oil residues to study its performance. SBS/REOB modified-rejuvenated asphalt was prepared by a high-speed shearing mechanism with aged asphalt, Recycled Engine Oil Bottom (REOB), Styrenic Block Copolymers (SBS) modifier, and stabilizer. The effects of SBS content, REOB content, shear time, and shear rate on the conventional physical properties of asphalt were studied by orthogonal grey correlation analysis, and the optimum preparation scheme of SBS/REOB modified-rejuvenated asphalt was determined. The high and low temperature rheological properties of SBS/REOB modified-rejuvenated asphalt were studied using the Multiple Stress Creep Recover (MSCR) test and bending beam rheological (BBR) test. The mechanism of SBS/REOB on the modification and regeneration of aged asphalt was explored through four component tests and Fourier transforms infrared spectroscopy. The results show that the optimum preparation scheme is 4.5% SBS dosage, 9% REOB dosage, 50~60 min shear time, and 4500 r/min shear rate. The addition of SBS improves the elastic recovery performance and high temperature deformation resistance of REOB rejuvenated asphalt. At the same time, the S-value decreases and the m-value increases, which significantly improves the low temperature cracking resistance of REOB rejuvenated asphalt. The addition of REOB achieves component blending and regeneration of aged asphalt by supplementing the light components. After the addition of SBS absorbs the light component and swelling reaction occurs, the whole modification-regeneration process is mainly physical co-mixing and co-compatibility.

## 1. Introduction

Asphalt regeneration refers to coordinating the components and improving the performance of aged asphalt by using new asphalt or regenerant with appropriate viscosity and cooperating with specific preparation methods. The performance indicators of rejuvenated asphalt at each stage are closely related to the dosage and type of regenerant [1,2,3]. Incorporating pellet with reclaimed asphalt pavement (RAP) is beneficial in improving the sustainability and cost-efficiency of the asphalt mixtures industry [4]. Some regenerant may induce some differences in one or more of the situations by encouraging the characteristics of asphalt [5]. Based on the principle of creative reuse of waste materials, domestic and foreign experts and scholars began to focus on researching and developing waste oil-based products as regenerating agents [6,7,8].

As early as 1992, Herrington [9] conducted a study using waste automotive motor oil residues that had undergone reduced pressure distillation as a stabilizing regenerant for road asphalt. Ackbarali D.S et al. [10] comparatively studied the mixture performance of waste motor oil rejuvenated asphalt and normal emulsified asphalt and found that the former has better road performance. Zhang Yan, Chen Meizhu et al. [11] studied the regeneration effect of waste engine oil and waste soybean oil on aged asphalt from macro and micro perspectives comparatively. The results showed that the optimal blending amounts of waste engine oil and waste soybean oil were 5% and 6%, respectively. The regeneration effect of waste engine oil was better than that of waste soybean oil.

Generally, 70–80% of recycled engine oil (REO) can be effectively rejuvenated by the aforementioned processes. However, the remaining residue (accounting for 20–30%) cannot be effectively recycled due to the presence of many impurities; this residue is ultimately called recycled engine oil bottom (REOB). Miaozhang Yu [12] compared the performance and content of each component of three REOB and regulated regenerants to verify the theoretical feasibility of REOB as a regenerant. Li Jin [13,14,15,16] et al. used REOB as an asphalt regenerant and investigated the high and low temperature performance of REOB rejuvenated asphalt using asphalt conventional performance test, dynamic shear rheology (DSR) test, and asphalt bending creep stiffness (BBR) test, and found that 7% of REOB could achieve complete regeneration of the three major indexes of aged asphalt, but the low temperature performance recovery was insufficient compared to the original sample base asphalt. It is usually challenging to return aged asphalt regeneration to the level of performance of the new base asphalt, while the use of composite modification can be more targeted to improve and optimize the asphalt performance within a controlled range, improving its road performance [17,18,19]. The further modification of the rejuvenated asphalt has also been an important research direction in the field of composite modified asphalt in recent years.

In summary, the use of suitable modifiers with REOB regenerator on the aged asphalt composite modification-regeneration, its environmental protection, and asphalt performance optimization has an important double significance—to achieve the “waste” into “treasure”. In this paper, an orthogonal test was designed to determine the preparation scheme of SBS/REOB modified-rejuvenated asphalt, and then to investigate its conventional properties, rheological properties, and microstructural composition to evaluate the improvement effect of SBS/REOB on the properties of aged asphalt.

## 2. Materials and Methods

### 2.1. Raw Materials

#### 2.1.1. Base Asphalt

The 70# A-grade road petroleum asphalt produced by Santobo Petrochemical Co., Ltd. (Binzhou, China), aged in the laboratory, as the raw asphalt for preparing SBS/REOB modified-rejuvenated asphalt. Based on the requirements of the <<Standard Test Methods of Bitumen and Bituminous Mixtures for Highway Engineering>>, the properties of the original base asphalt were measured as shown in Table 1.

#### 2.1.2. REOB Regenerant

The REOB selected for this test came from a waste engine oil treatment plant in Zibo, Shandong Province, China, whose primary treatment process was: waste engine oil filtration-thin film distillation-sugar aldehyde refining-white clay process. From the physical and chemical properties in Table 2, the REOB meets the specification of a conventional RA-1 regenerant.

#### 2.1.3. SBS Modifier

The modifier was adopted from SBS1401 (YH-792), a thermoplastic styrene-butadiene rubber produced by Yueyang Baling Petrochemical (Yueyang, China), which is linear SBS with an S/B block ratio of 40/60.

#### 2.1.4. Stabilizer

The active ingredient of the asphalt stabilizer used was sulfur, with a content of 99.9% or more, a moisture content of 0.02%, a melting point of 120 °C, and an 80-mesh sieve margin of 0.1%.

#### 2.1.5. Preparation of the Aged Asphalt Binders

Aged asphalt preparation method: we selected 40 cm × 30 cm × 4.8 cm stainless steel plates, 600 g of new asphalt was poured into each plate. The asphalt had a plate thickness of about 0.5 cm, and was put into the 163 °C oven heating insulation for 48 h, and stirred every 2 h to ensure that the material is uniformly heated to prevent the occurrence of surface-crusting phenomenon. The performance test results are shown in Table 3.

Table 3 showed that the 48 h oven-aged asphalt was more heavily aged than PAV-aged asphalt, and the performance of “48 h oven” aged asphalt was between “5 years of service” and “10 years of service” compared to the extracted and rejuvenated asphalt. The performance of “48 h oven” aged asphalt was between that of “5 years’ service” and “10 years’ service” recovered asphalt.

### 2.2. Design of Orthogonal Test

We set three levels, respectively, designed a L9(3^4^) orthogonal test, as shown in Table 4, and prepared 9 groups of SBS/REOB modified-rejuvenated asphalt under different conditions.

### 2.3. Preparation Process

(a)REOB was added to the thermally aged asphalt, and a specific mass fraction of REOB (factor A) was dispersed into the asphalt at 150 °C using a mixer at 2500 r/min for 20 min.(b)Set the shear speed to 3000 r/min and add the required SBS (factor B) within 5 min. Control the temperature to 190 °C.(c)After shearing at a certain speed (factor D) for a certain time (factor C), add 6% stabilizer of SBS dosage, and then stir at a low speed of 1000 r/min for 10 min to remove air bubbles.(d)Dissolution development at 160 °C for 1 h.

### 2.4. Experimental Method

#### 2.4.1. Routine Performance Test

Nine groups of modified-rejuvenated asphalt were tested for penetration, softening point, ductility, and elastic recovery rate under the orthogonal test scheme. The results were analyzed to obtain the optimal dosing values of SBS and REOB as well as the optimal shear time and shear rate.

#### 2.4.2. High-Temperature Rheological Property Test

The Multiple Stress Creep Recovery test (MSCR) used the delayed elastic recovery performance of asphalt under applied stress to evaluate the high-temperature performance of the binder, and the cumulative strain of asphalt had a good correlation with the high-temperature performance of asphalt [20,21,22].

Referring to the American Association of State Highway and Transportation Officials (AASHTO) test protocol, MSCR tests were conducted using a dynamic shear rheometer (DSR) on SBS/REOB modified-rejuvenated asphalt and SBS modified asphalt and REOB-rejuvenated asphalt with the same admixture dose under optimal preparation conditions. The test temperature was set at 64 °C. According to the loading stress, the test was divided into two stages of 0.1 kPa and 3.2 kPa stress loading, and each step was loaded in the form of 10 cycles, 10 s/cycle (loading lasts 1 s, unloading lasts 9 s) to obtain the creep recovery rate R and irrecoverable creep flexibility J_nr_ for each asphalt [23,24].

#### 2.4.3. Low-Temperature Rheological Property Test

Standard size 101.6 mm × 12.7 mm × 6.4 mm beam specimens of PAV aged asphalt was prepared and tested by Bending Beam Rheometer (BBR) at −6 °C, −12 °C, and −18 °C for the bending creep modulus S and slope m of the creep curve.

Among the S and m results of the 8th, 15 s, 30 s, 60 s, 120 s and 240 s automatically tested by the computer, the test result at 60 s was taken as the final value.

#### 2.4.4. Microscopic Test

The four components of asphalt were tested and analyzed using rod-thin-layer chromatography/hydrogen flame ionization detection (TLC-FID) and the functional group changes were analyzed using infrared spectroscopy to reveal the modification-regeneration mechanism of SBS/REOB on aged asphalt [25,26].

## 3. Results and Discussion

### 3.1. Determination of the Best Preparation Conditions

The sample preparation and sample performance tests were carried out in groups according to the orthogonal design, and the physical property test results of each group of asphalt samples are shown in Table 5.

Based on the experimental results, the data were processed using grey correlation analysis with the following procedure:(a)The matrix sequences are listed in Table 5 Orthogonal test results. Here, the mean value of each group of indicators is taken as the reference sequence, i.e., X0 = (62.10, 63.40, 58.75, 60.38, 64.03, 61.15, 59.75, 66.10, 58.35).(b)Dimensionless treatment of the index series(c)Solving the difference series with two polar differences(d)Calculate the correlation coefficient, degree, and proportion of factors.(e)Results processing: A weighted average score is given to each indicator factor based on obtaining the proportion of each factor. We call this method the grey correlation composite scoring method.

Let the difference between the maximum value and the minimum value of the test result of each test index be *h, b_ij_* = the proportion of each factor/h, the score value is Fi = $\overset{4}{\underset{j=1}{\sum }}b_{ij}\times test\;index\;value$. Since the smaller the penetration index value is, the better the viscosity is, the value of this index was taken as negative in the scoring calculation here.

The obtained scoring results are processed for polar difference calculation, and the method is called the gray correlation polar difference method. The obtained scores and the results of the polar difference calculation are shown in Table 6. 

(f) Analysis of results.

REOB content was the most important factor affecting the performance of asphalt samples, followed by SBS content, which had the greatest influence on the penetration index. The shear time and shear rate mainly affected the elastic recovery performance of materials and had the least influence on the softening point index. The degree of influence of each factor on the product effect is REOB dosage > SBS dosage > shear time > shear rate.

Based on the evaluation principle that the higher the overall score is, the better the product performance, the highest score of 489.21 was obtained for the A2-B3-C1-D2 solution, i.e., ”9% REOB, 4.5% SBS dosage, 50 min shear at 4500 r/min” was the preferred preparation condition. The results of the grey correlation ANOVA showed that the preferred solution was A2-B3-C1-D3, i.e., “9% REOB, 4.5% SBS dosage, shear at 4500 r/min for 60 min”. The difference between them only lies in the shearing time, so the optimal preparation condition is recommended as “9% mass fraction of REOB + 4.5% SBS, shearing 50~60 min at 4500 r/min”.

### 3.2. High-Temperature Rheological Properties

The SBS/REOB modified-rejuvenated asphalt under two processes was selected as the main object of the study. The MSCR test was carried out in comparison with the as-built base asphalt, aged asphalt, and SBS modified asphalt. Figure 1 and Figure 2 show the MSCR test results for each asphalt at 0.1 kPa and 3.2 kPa stress levels, respectively.

From Figure 1 and Figure 2, it can be seen that the strains generated by 9% REOB + aged asphalt were at the highest level of all stress levels. The peak strains at 0.1 kPa and 3.2 kPa were 12 and 15 times higher than those of SBS modified asphalt, which shows that the SBS modification of REOB rejuvenated asphalt can effectively improve its stability performance under externally applied loads. The difference between the strain values of SBS/REOB modified-rejuvenated asphalt and REOB modified asphalt becomes more prominent as the load becomes stronger, i.e., the resistance to load deformation of the modified-rejuvenated asphalt is more prominent at high stress levels. From the comparison of the strain values generated by the three SBS modified asphalts, 4.5% SBS + 9% REOB + 50 min + aged asphalt produced the smallest strain value and was better than 4.5% SBS + base asphalt, which had the best high temperature resistance to deformation.

Additionally, creep recovery rate R, % and irrecoverable creep flexibility J_nr_, kPa^−1^ are usually used to evaluate the high-temperature creep recovery performance of asphalt to reflect the high temperature anti-rutting performance. Practical pavement applications with a comparative index of stress sensitivity J_nr,diff_ at 3.2 kPa versus 0.1 kPa.

Table 7 gives the creep recovery rates R_0.1_, R_3.2_, and irrecoverable creep flexibility J_nr, 0.1_, J_nr, 3.2_ for the above four asphalt bonds at 0.1 kPa and 3.2 kPa stress levels, respectively, according to the results plotted in Figure 3 and Figure 4.

Figure 3 combined with Table 7 test results shows that the R of 9% REOB + aged asphalt under all stress levels is more than 1 times higher than that of aged asphalt but cannot recover to the level of base asphalt. Under 0.1 kPa stress level, the modified-aged asphalt R value can be more than 50%, compared to REOB rejuvenated asphalt creep recovery performance improved by more than 16 times. Overall, 3.2kPa under the R reached more than 30%, and 4.5S + 9R + 50 + A can achieve 4.5S + B of its R value of 93.3%. This indicates that SBS makes the REOB rejuvenated asphalt binder in the proportion of elasticity increases, so that the state of the binder is more elastomeric, SBS on rejuvenated asphalt “elasticity” effect is obvious, so the elastic recovery performance is improved, but its “elasticity” ability is limited, and cannot fully reach the level of ordinary SBS-modified asphalt. The sensitivity of different materials to high stress is different, and the base asphalt and 9% REOB rejuvenated asphalt have the highest R_0.1_:R_3.2_ values, 5.9 and 10.1, respectively, while the R_0.1_:R_3.2_ of modified-rejuvenated asphalt is not more than 2, more “high temperature”, “heavy load”, under the resistance of deformation, with 4.5S + 9R + 50 + A best.

As can be seen from Figure 4, the J_nr_ values of modified-rejuvenated asphalt are significantly lower than those of base asphalt and rejuvenated asphalt. At a high stress level of 3.2 kPa, the J_nr_ values of 4.5S + 9R + 50 + A and 4.5S + 9R + 60 + A decreased by 94.0% and 95.6%, respectively, compared with those of REOB rejuvenated asphalt, which shows that the J_nr_ values of 4.5S + 9R + 60 + A modified-rejuvenated asphalt are the best. This result indicates that the 4.5S + 9R + 60 + A modified-rejuvenated asphalt has the best high temperature deformation resistance. In addition to the aged asphalt, the J_nr,diff_ values of 4.5S + B were the smallest, 4.5S + 9R + 50 + A was the second smallest, less than 25%, and the J_nr,diff_ values of B and 9R + A were more than 30%, indicating that the modified-rejuvenated asphalt materials prepared under this process are less sensitive to stress changes, which is closely related to the good stability of SBS modifier. The J_nr,diff_ value of asphalt binder has excellent stress sensitivity at less than 75%, which can make the asphalt pavement have better rutting resistance, and the materials prepared by the modification-rejuvenated process in this study all meet this requirement.

The differences between R and J_nr_ values of modified-rejuvenated asphalt obtained at different shear times are small, with 4.5S + 9R + 50 + A showing more excellent elastic recovery performance and stress sensitivity, and 4.5S + 9R + 60 + A having more advantages in high temperature deformation resistance.

### 3.3. Low-Temperature Rheological Properties

The US SHRP specification states that for PAV-aged asphalt, the stiffness modulus value should be less than 300 MPa at 60 s loading and the creep rate m should be not less than 0.30. From the BBR test results of each asphalt at different temperatures in Figure 5, it can be seen that the S and m values of the aged asphalt at −12 °C are far above this specification, so the aging asphalt is not continued to be tested at lower temperatures. Therefore, the test is not continued at lower temperatures.

As shown in Figure 5, comparing the S-value results of different materials at each temperature, it can be seen that the S-value of 9% REOB rejuvenated asphalt 9R + A is significantly lower and the m-value is significantly higher than that of aged asphalt, indicating that the use of REOB on aged recycled can improve the low-temperature performance of asphalt, but does not reach the low-temperature performance level of base asphalt B. SBS/REOB modified-rejuvenated asphalt S-value and lower than B with 9R + A, m-value increased, indicating that modified-rejuvenated asphalt relative to rejuvenated asphalt low-temperature performance to obtain further improvement, and better than the base asphalt.

At −12 °C and −18 °C test temperatures, the S and m values of the two modified-rejuvenated asphalt 4.5S + 9R + 50 + A and 4.5S + 9R + 60 + A prepared at different shear times can be basically achieved to the level of ordinary SBS modified asphalt 4.5S + B. At −24 °C, the m of 4.5S + 9R + 60 + A cannot meet the SHRP specification, and its low temperature cracking resistance is not as good as 4.5S + 9R + 50 + A. The reason is that the excessive shear time leads to different degrees of aggregation of SBS particles, which destroys the composition of the SBS modified asphalt network structure, or excessive shear makes the SBS polymer chain break, and the asphalt has partially aged. Therefore, from the viewpoint of low temperature performance, 50 min is recommended as the optimal shear time for SBS/REOB modified-rejuvenated asphalt preparation.

### 3.4. Micro-Mechanism Analysis

#### 3.4.1. Four Components

Aged asphalt, 70# base asphalt, 9% REOB rejuvenated asphalt, 4.5% SBS modified asphalt, and 4.5% SBS + 9% REOB modified-rejuvenated asphalt were selected for four-component analysis, and the results are shown in Figure 6.

(1)As shown in Figure 6, compared with the base asphalt, the change amounts of asphaltenes resins, aromatics, and saturates in the aged asphalt are 7.6, 21.3, −22.2, and −7.7, respectively, which means that the saturates and aromatics of light components are reduced, and the asphaltene and resins of a recombinant fraction are significantly increased, and component migration occurs. The change of four groups of rejuvenated asphalt compared with aging asphalt is −7.4, −22.6, 20.3, 8.7, the light component increases, and aging asphalt light component loss, which indicates that the high proportion of aromatics contained in REOB can supplement the asphalt in the aging process of the lack of aromatics part, so that the light component content increased, the recombination fraction decreased, to achieve asphalt regeneration, the regeneration has restored the composition of the asphalt components that had changed during the aging process, which resulted in macroscopic performance recovery.

Based on the blending component theory, assuming that REOB regenerates the aged asphalt and its mechanism is pure component blending, the content of each component of the rejuvenated asphalt should satisfy the calculation result of Equation (1).
P = m_a_P_a_ + m_r_P_r_,(1)
where P is the proportion of a component in REOB rejuvenated asphalt (%); P_a_ is the proportion of a component in aged asphalt (%); P_r_ is the proportion of a component in REOB (%); m_a_ is the blending ratio of aged asphalt in rejuvenated asphalt; m_r_ is the blending ratio of REOB in rejuvenated asphalt.

The results of the measured and calculated values of the four components of 9% REOB rejuvenated asphalt are shown in Table 8.

Comparing the measured values with the theoretical calculated values, it is found that there are apparent differences between them. Therefore, the regeneration process is not a pure component reconciliation theory, and there are physical or chemical reactions and migration among the four components that cannot be ignored.

(2)The SBS modification of base asphalt and REOB asphalt also caused the redistribution of the components in asphalt. The proportion of each component is reduced in the light component and increased in the recombination component, in which the light component is mostly aromatics and a small part of saturates, which corresponds to the macroscopic properties of ductility, softening point increase, and penetration decrease, further verifying from the microscopic chemical components that the modified-renewed asphalt is improved by SBS absorption of light component swelling reaction. The same amount of SBS is used in the modified-rejuvenated asphalt, which is less migratory than the directly modified asphalt, indicating that it is more difficult to modify the rejuvenated asphalt, so its performance improvement is limited.

#### 3.4.2. Infrared Spectrum

The transmittance-wave number infrared absorption spectra (FTIR) plots of the regenerant REOB with modified asphalt, rejuvenated asphalt, and modified-rejuvenated asphalt with base asphalt are given in Figure 7 and Figure 8, respectively.

From Figure 7, it can be seen that the REOB showed sharp strong absorption peaks at 2929.7 cm^−1^ and 2856.3 cm^−1^, which were generated by CH_2_ asymmetric vibration and symmetric vibration, respectively, indicating that the regenerant contains non-polar methylene; the asymmetric bending vibration peak of CH_3_ appeared at 1439.6 cm^−1^, and the generation of these characteristic peaks indicates that the REOB contains a higher proportion of saturated hydrocarbons. A strong absorption peak appears at 703 cm^−1^ in the fingerprint area, resulting from the vibration of the outer C=H bending surface of the aromatic ring, indicating that the REOB contains light components of aromatic hydrocarbons. In summary, it can be inferred that REOB mainly comprises alkanes and saturated hydrocarbons such as cycloalkanes and aromatic compounds.

Figure 8 shows that the positions of the peaks appearing in different asphalt are generally consistent, and there are differences in the intensity of the absorption peaks. The four asphalts have broad and weak absorption peaks at 3444.7 cm^−1^, which are considered to be generated by N-H bonding vibrations or O-H bonding vibrations in alcohols and phenols. The absorption peaks at 2927.8 cm^−1^ and 2854.5 cm^−1^ are generated by methylene CH_2_ vibrations, while 1461.9 cm^−1^ and 1384.8 cm^−1^ show characteristic absorption peaks generated by methyl CH_3_ bending vibrations. 4.5S + 9R + 50 + A modified-rejuvenated asphalt has a characteristic peak at 866.0 cm^−1^ caused by =C-H-bonds on the benzene ring, which indicates the presence of benzene ring aromatic hydrocarbons in the tested material.

Both 4.5S + B and 4.5S + 9R + 50 + A show distinct SBS characteristic peaks at 966.3 cm^−1^ and 696.5 cm^−1^, generated by C=C bond distortion and C-H bond vibration in the benzene ring, respectively, where the C=C bond is present in the polybutadiene of SBS and the C-H bond is present in the polystyrene segment. This result indicates that the modified-rejuvenated asphalt obtained is consistent with the conventional modified asphalt in terms of composition.

9R + A and 4.5S + 9R + 50 + A have more substantial aromatic absorption peaks at 866.0 cm^−1^ than B and 4.5S + B. In comparison, B and 4.5S + B have higher intensity alkyl absorption peaks, which means that 9% REOB can replenish the lost saturates, aromatics, and other components in the aging asphalt, i.e., the mutual migration and transformation of functional groups, which is eventually reflected in the recovery of asphalt properties. However, there is a limit to the regeneration ability.

In summary, the composition of the chemical components of asphalt mainly includes alkanes, cycloalkanes, aromatic compounds and heteroatomic derivatives, etc. The composition of REOB regenerant is similar to that of REOB, thus proving the regeneration effect of REOB on aging asphalt from the perspective of the composition. However, the results of the four-component analysis do not exclude a weak chemical reaction. The IR spectrum of the asphalt modified with SBS for REOB, i.e., 4.5% SBS + 9% REOB, shows only the new characteristic peaks of SBS, which is a simple superposition of SBS modifier and REOB rejuvenated asphalt, and the process involves physical co-mixing and co-compatibility.

## 4. Conclusions

(1)The degree of influence of SBS/REOB modified-rejuvenated asphalt performance using grey correlation analysis is ranked as: REOB dosage > SBS dosage > shear time > shear rate. The recommended vital preparation parameters are “9% REOB + 4.5% SBS dosage, shear at 4500 r/min for 50 min~60 min”.(2)The addition of SBS modifier helps to improve the creep recovery rate R and reduce the irrecoverable creep flexibility J_nr_ and J_nr, diff_ of REOB rejuvenated asphalt, so that the obtained SBS/REOB modified-rejuvenated asphalt has higher elastic recovery performance, higher temperature deformation resistance, and lower stress sensitivity. SBS/REOB modified-rejuvenated asphalt has a lower S value and higher m value than REOB rejuvenated asphalt, which has better low-temperature ductility and flexibility.(3)The four-component and FTIR tests show that the regeneration of aged asphalt by REOB is a non-complete component reconciliation, and the presence and location of the characteristic peaks of modified-rejuvenated asphalt is a simple superposition of SBS modifier and REOB rejuvenated asphalt. The modification-regeneration mechanism of SBS/REOB on aged asphalt is physically dominated, accompanied by weak chemical reactions.

## Figures and Tables

**Figure 1 polymers-14-05071-f001:**
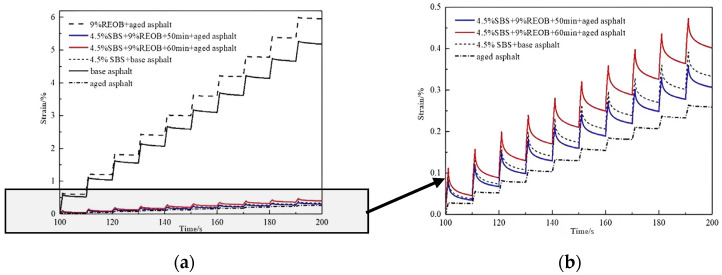
MSCR test results at 0.1 kPa stress level: (**a**) Test results of each asphalt; (**b**) Test results of aged asphalt, 4.5% SBS + 9% REOB + 50 min + aged asphalt, 4.5% SBS + 9% REOB + 60 min + aged asphalt and 4.5% SBS + base asphalt.

**Figure 2 polymers-14-05071-f002:**
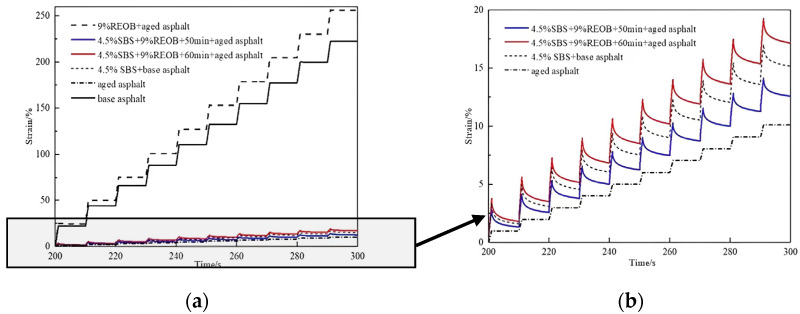
MSCR test results at 3.2 kPa stress level: (**a**) Test results of each asphalt; (**b**) Test results of aged asphalt, 4.5% SBS + 9% REOB + 50 min + aged asphalt, 4.5% SBS + 9% REOB + 60 min + aged asphalt, and 4.5% SBS + base asphalt.

**Figure 3 polymers-14-05071-f003:**
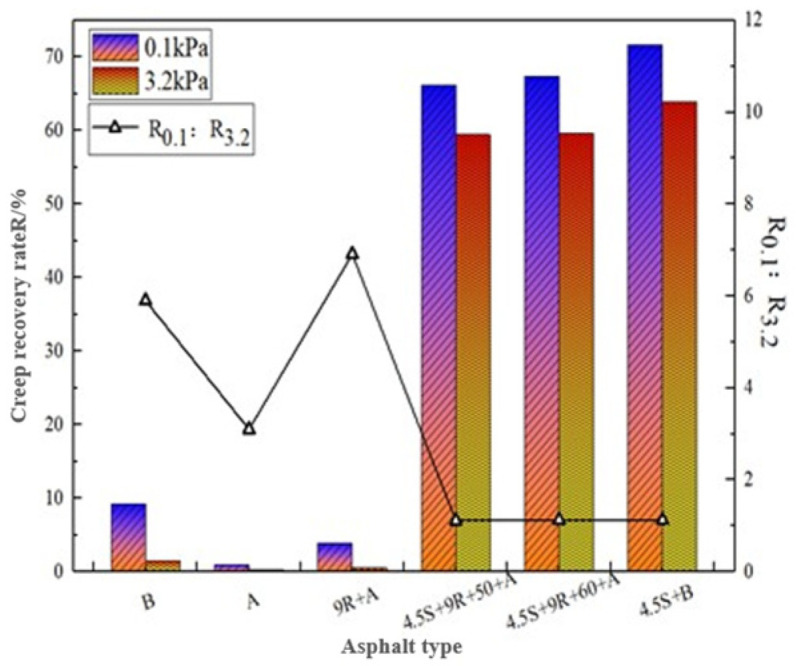
Comparison of R for different asphalt types in the Figure.

**Figure 4 polymers-14-05071-f004:**
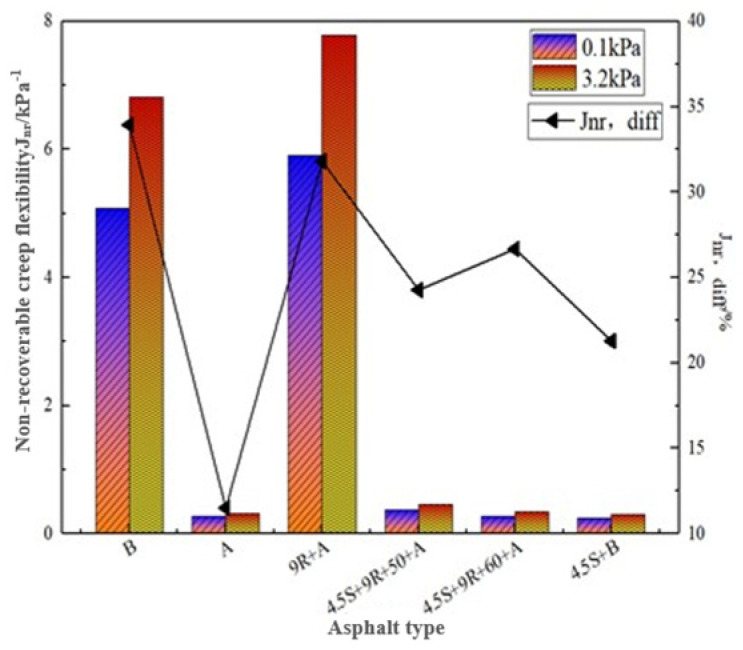
Comparison of J_nr_ for different asphalt types.

**Figure 5 polymers-14-05071-f005:**
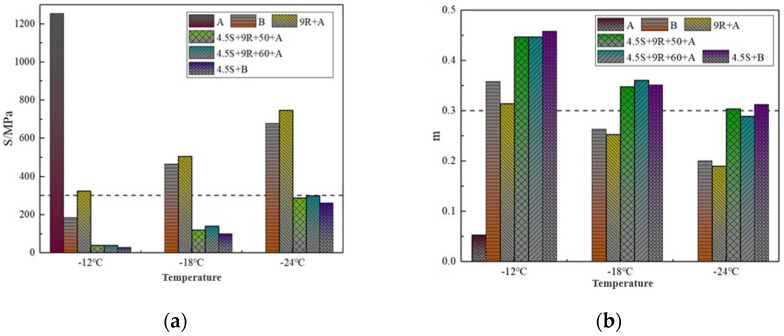
BBR test results of each asphalt at different temperatures: (**a**) Modulus of rigidity; (**b**) Creep rate.

**Figure 6 polymers-14-05071-f006:**
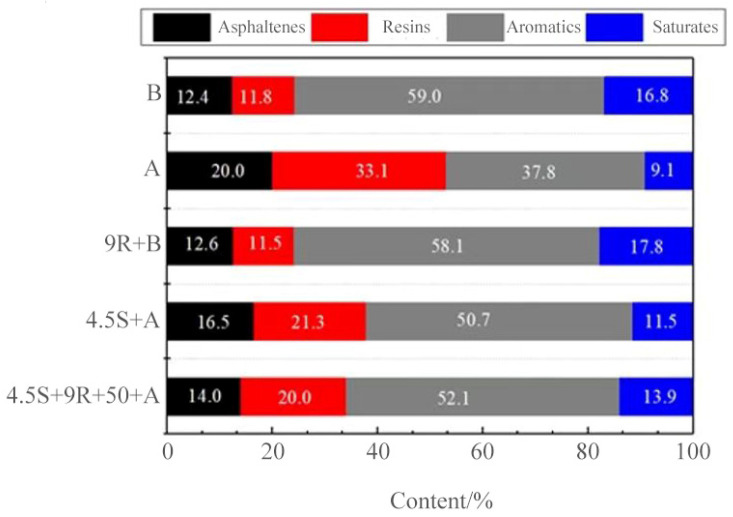
Percentage of four components of different types of asphalt.

**Figure 7 polymers-14-05071-f007:**
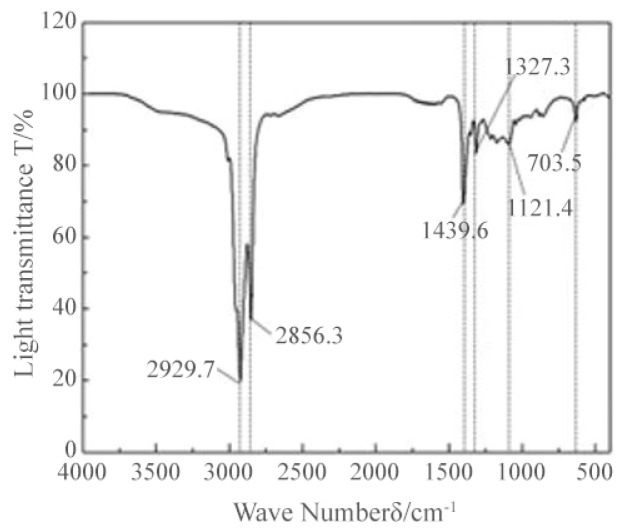
IR spectrum of REOB.

**Figure 8 polymers-14-05071-f008:**
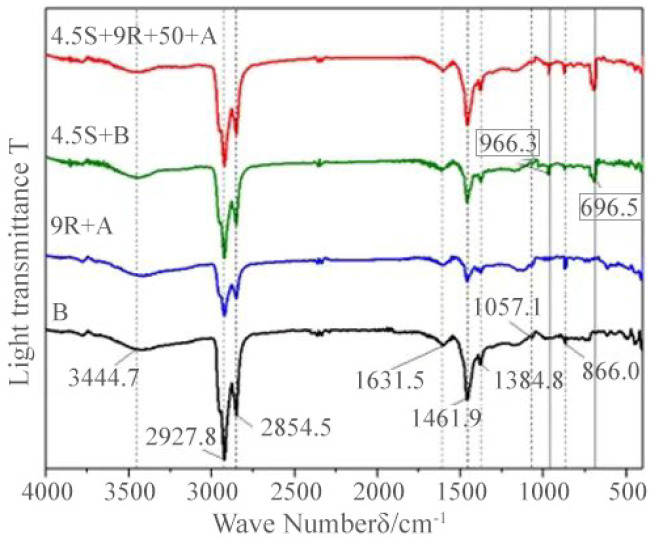
Infrared spectra of different types of asphalt.

**Table 1 polymers-14-05071-t001:** Performance index of 70# base asphalt.

Test Project		Test Results	Standard Indicators
Needle penetration 100 g, 5 s, 25 °C/0.1 mm		73.2	60–80
Insertion index PI		−1.1	−1.5~1.0
Softening point/°C		49.7	≥46
60 °C dynamic viscosity/Pa·s		271	≥180
Viscosity (135 °C) /Pa·s		0.266	
10 °C degrees/cm		73.0	≥25
15 °C degrees/cm		>100	≥100
Flash point (COC)/°C		280	≥260
TFOT (Thin Film) Oven Test) after 163 °C, 5 h	Quality change/%	−0.255	≤±0.8
	Residual penetration ratio (25 °C)	68.3	≥61
	Residual penetration ratio (25 °C)	10.6	≥6

**Table 2 polymers-14-05071-t002:** Physicochemical properties of REOB.

Inspection Items		R-1	RA-1 Regenerant
Density/(g/cm^3^)		0.911	The measured
Flash point/°C		258	≥220
Aromatic content/%		83.1	The measured
Saturation fraction content/%		4.6	≤30
TFOT post	Viscosity ratio	1.29	≤3
	Quality change /%	−1.89	−4~4

**Table 3 polymers-14-05071-t003:** Comparison of asphalt performance indexes under different aging methods.

Test Project	48 h Oven Aging	PAV	5 Years of Recycling Asphalt	10 Years of Recycling Asphalt
Penetration (25 °C)/mm	22.0	26.6	24.6	17.6
Softening point/°C	67.6	62.3	64.9	76.6
Ductility (10 °C, 5 cm/min)/cm	0.2	1.7	2.1	brittle break
Viscosity (135 °C) /Pa·s	2.247	1.967	2.118	2.763

**Table 4 polymers-14-05071-t004:** Orthogonal test design scheme.

Test Group	Horizontal Combinations	Test Factor			
		A	B	C	D
		REOB Dosage /%	SBS Dosage /%	Shearing Time/min	Shear Rate /(r/min)
1	A1B1C1D1	1 (11)	1 (3.5)	1 (40)	1 (3500)
2	A1B2C2D2	1 (11)	2 (4.0)	2 (50)	2 (4500)
3	A1B3C3D3	1 (11)	3 (4.5)	3 (60)	3 (5500)
4	A2B1C3D2	2 (9)	1 (3.5)	2 (50)	3 (5500)
5	A2B2C3D1	2 (9)	2 (4.0)	3 (60)	1 (3500)
6	A2B3C1D2	2 (9)	3 (4.5)	1 (40)	2 (4500)
7	A3B1C3D2	3 (7)	1 (3.5)	3 (60)	2 (4500)
8	A3B2C1D3	3 (7)	2 (4.0)	1 (40)	3 (5500)
9	A3B3C2D1	3 (7)	3 (4.5)	2 (50)	1 (3500)

**Table 5 polymers-14-05071-t005:** Results of the orthogonal test.

Test Group	Penetration/mm	Softening Point/°C	Ductility/cm	Elastic Recovery/%
1	84	57.5	23.9	81
2	80.2	61.2	25.2	84
3	63.8	63.9	29.3	83
4	78.3	55.7	26.5	79
5	76	62.3	27.2	88
6	64.5	63.7	33.4	86
7	76.9	58.3	26.8	77
8	79.1	64.1	29.8	89
9	62.7	65.2	30.5	81

**Table 6 polymers-14-05071-t006:** Gray correlation composite score and extreme difference analysis results.

Horizontal Combinations	Test Factor				Rating Fi
	A	B	C	D	
	REOB Dosage /%	SBS Dosage/%	Shearing Time/min	Shear Rate/(r/min)	
A1B1C1D1	1 (11)	1 (3.5)	1 (40)	1 (3500)	313.73
A1B2C2D2	1 (11)	2 (4.0)	2 (50)	2 (4500)	338.18
A1B3C3D3	1 (11)	3 (4.5)	3 (60)	3 (5500)	370.38
A2B1C3D2	2 (9)	1 (3.5)	2 (50)	3 (5500)	408.58
A2B2C3D1	2 (9)	2 (4.0)	3 (60)	1 (3500)	459.61
A2B3C1D2	2 (9)	3 (4.5)	1 (40)	2 (4500)	489.21
A3B1C3D2	3 (7)	1 (3.5)	3 (60)	2 (4500)	409.39
A3B2C1D3	3 (7)	2 (4.0)	1 (40)	3 (5500)	470.65
A3B3C2D1	3 (7)	3 (4.5)	2 (50)	1 (3500)	475.26
$\bar{K_{1}}$	340.76	377.23	425.53	416.20	Factor Priorities A > B > C > D
$\bar{K_{2}}$	452.77	423.81	407.34	412.26	
$\bar{K_{3}}$	451.77	444.95	413.13	417.54	
Range R	111.70	67.72	17.19	4.28	
Optimum	A2	B3	C1	D3	

**Table 7 polymers-14-05071-t007:** Calculated results of MSCR tests for each asphalt at different stress levels.

Number	Asphalt Type	R_0.1_/%	R_3.2_/%	J_nr,0.1_/ kPa^−1^	J_nr,3.2_/ kPa^−1^	J_nr,diff_/%
B	Base asphalt	9.26	1.57	5.085	6.811	33.94
A	Aged asphalt	0.93	0.30	0.278	0.310	11.51
9R + A	9% REOB + aged asphalt	3.94	0.57	5.909	7.788	31.80
4.5S + 9R + 50 + A	4.5% SBS + 9% REOB + 50 min + aged asphalt	67.34	59.65	0.375	0.466	24.27
4.5S + 9R + 60 + A	4.5% SBS + 9% REOB + 60 min + aged asphalt	66.22	59.46	0.270	0.342	26.67
4.5S + B	4.5% SBS + base asphalt	71.71	63.92	0.249	0.302	21.28

**Table 8 polymers-14-05071-t008:** Calculated results of the four components of 9% REOB.

Object		Asphaltenes/%	Resins/%	Aromatics/%	Saturates/%
Aged asphalt		20.0	33.1	37.8	9.1
REOB		3.6	8.7	83.1	4.6
9% REOB	Measured value	12.6	11.5	58.1	17.8
	Calculated values	18.6	31.1	41.5	8.7

## Data Availability

The experimental data in this paper are from the pavement material laboratory of Shandong Jiaotong University, which is the provincial key laboratory.

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
