# Peer review of "Preparation and Macro-Micro Properties of SBS/REOB Modified-Rejuvenated Asphalt"

_polymers, 2022, doi:10.3390/polym14235071_

Round 1
Reviewer 1 Report
Generally, the paper looks a test report rather than an academic paper. The impetus of paper/study is not clear. The following issues should have been addressed during the paper revision:
1) What are the gaps in the extant literature that triggered the authors to conduct the current research?
2) What are the novelties of the current research? I just wonder how the research or paper can be positioned in the relevant literature.
3) What are the "practical" implications of the research findings?
4) The paper has a lot of grammatical errors. The authors should have the paper proofread by a professional English writer before submission.
5) What are the limitations of the research? How do these limitations affect the interpretation of the research findings?
6) The methodology of data collection has not been well-explained. For example, how many samples were taken for each group?
Author Response
Dear Professor,
I am very pleased to receive the letter from the reviewers' review of the manuscript of " Preparation and macro-micro properties of SBS/REOB Modi-fied- Rejuvenated Asphalt" (ID: polymers-2035518). Thank you for your comments on our paper. These review comments are of high value to the revision and improvement of our manuscript, and also have important guiding significance for our research work. After carefully studying these comments and your advice, we have supplemented our analyses, and revised the manuscript content. Our responses to the comments are provided below.
Please refer to the attachment for details of the revised version
Point 1: What are the gaps in the extant literature that triggered the authors to conduct the current research?
Response 1: Thanks to the reviewers for their suggestions. The advantages of REOB modification are discussed in the existing literature, but the disadvantages of REOB are studied and corrected in this paper, which can make more effective use of REOB.
Point 2: What are the novelties of the current research? I just wonder how the research or paper can be positioned in the relevant literature.
Response 2: Thanks to the reviewers for pointing out the problem. The advantages of REOB modification are discussed in the existing literature, but the disadvantages of REOB are studied and corrected in this paper, which can make more effective use of REOB.the original intention of this paper is to focus on the shortcomings of REOB rejuvenated asphalt, and try to use SBS to improve it.
Point 3: What are the "practical" implications of the research findings?
Response 3: Thanks to the reviewer for the correction. . The advantages of REOB modification are discussed in the existing literature, but the disadvantages of REOB are studied and corrected in this paper, which can make more effective use of REOB.the original intention of this paper is to focus on the shortcomings of REOB rejuvenated asphalt, and try to use SBS to improve it.In this paper, we designed orthogonal tests to prepare SBS/REOB modified-recycled asphalt, and studied its conventional properties, rheological properties and microstructural composition to evaluate the improvement effect of SBS/REOB on the properties of aged asphalt.
Point 4: The paper has a lot of grammatical errors. The authors should have the paper proofread by a professional English writer before submission.
Response 4: Thank you for your comments, The grammatical errors in this article have been carefully corrected
Point 5: What are the limitations of the research? How do these limitations affect the interpretation of the research findings?
Response 5: Thanks to reviewers for their suggestions. The advantages of REOB modification are discussed in the existing literature, but the disadvantages of REOB are studied and corrected in this paper, which can make more effective use of REOB.the original intention of this paper is to focus on the shortcomings of REOB rejuvenated asphalt, and try to use SBS to improve it.In this paper, we designed orthogonal tests to prepare SBS/REOB modified-recycled asphalt, and studied its conventional properties, rheological properties and microstructural composition to evaluate the improvement effect of SBS/REOB on the properties of aged asphalt.The limitation is that it is expensive to apply in the field and cannot be widely used in practical applications. The limitation does not have much influence on the study results, but still we need to control the cost.
Point 6: The methodology of data collection has not been well-explained. For example, how many samples were taken for
Response 6: Thanks to reviewers for pointing out the problem. The large number of experiments in this paper does not allow for a large space to present the sample size, and a sufficient number of experiments were used for each group.

Reviewer 2 Report
11. Abstract cannot be starting with methodology. “SBS/REOB modified-rejuvenated asphalt was prepared by….”
22. Abstract not enough information. Background study, problem statement, points out research gaps, aims & objectives, summary of methods, and novelty of research study are not clearly presented.
33. Introduction lack of information on related study. To improve these, suggest review and add as follow: (a). https://doi.org/10.1007/s13369-021-05413-0; (b). doi:10.1088/1755-1315/682/1/012071.
44. "Asphalt regeneration refers to the process of coordinating the components and improving the performance…”. Supporting references are needed (https://doi.org/10.15282/construction.v1i1.6324 and https://doi.org/10.15282/construction.v1i1.6282).
55. Table 2.1 to 2.4. More descriptions and discussions are required. Not only data presented. In addition, what the standard use for the test project? ASTM? BS EN? And etc.
66. Table 3.1 & 3.2. More discussions are compulsory. Not only data presented.
77. For reputable journal, minimum references are 20.
Author Response
Dear Professor,
I am very pleased to receive the letter from the reviewers' review of the manuscript of " Preparation and macro-micro properties of SBS/REOB Modi-fied- Rejuvenated Asphalt" (ID: polymers-2035518). Thank you for your comments on our paper. These review comments are of high value to the revision and improvement of our manuscript, and also have important guiding significance for our research work. After carefully studying these comments and your advice, we have supplemented our analyses, and revised the manuscript content. Our responses to the comments are provided below.
Please refer to the attachment for details of the revised version
Point 1: Abstract cannot be starting with methodology. “SBS/REOB modified-rejuvenated asphalt was prepared by….”
Response 1: Thanks to the reviewers for their suggestions. This question has been changed in the text at page 1 line 9.
Point 2: Abstract not enough information. Background study, problem statement, points out research gaps, aims & objectives, summary of methods, and novelty of research study are not clearly presented.
Response 2: Thanks to the reviewers for pointing out the problem.
Point 3: Introduction lack of information on related study. To improve these, suggest review and add as follow: (a). https://doi.org/10.1007/s13369-021-05413-0; (b). doi:10.1088/1755-1315/682/1/012071.
Response 3: Thanks to the reviewer for the correction. Relevant literature has been included in the text at page 1 line 36.
Point 4: "Asphalt regeneration refers to the process of coordinating the components and improving the performance…”. Supporting references are needed (https://doi.org/10.15282/construction.v1i1.6324 and https://doi.org/10.15282/construction.v1i1.6282).
Response 4: Thank you for your comments, Relevant literature has been included in the text at page 1 line 38 and page 2 line 57.
Point 5: Table 2.1 to 2.4. More descriptions and discussions are required. Not only data presented. In addition, what the standard use for the test project? ASTM? BS EN? And etc.
Response 5: Thanks to reviewers for their suggestions. The reference standard is 《Standard Test Methods of Bitumen and Bituminous Mixtures for Highway Engineering》, located at page 2 line 84 in the text
Point 6: Table 3.1 & 3.2. More discussions are compulsory. Not only data presented.
Response 6: Thanks to reviewers for pointing out the problem. Having listened to your suggestions, there is more discussion of Tables 3.1 and 3.2 at page 6 line 189.
Point 7: For reputable journal, minimum references are 20.
Response 7: Thank you for your comments, The references are now 21.

Reviewer 3 Report
Review of the manuscript (Manuscript ID: polymers-2035518): “Preparation and macro-micro properties of SBS/REOB Modified- Rejuvenated Asphalt” submitted for publication in the journal Polymers.
The results reported in this article are interesting and valuable for the experts working in the field of road asphalt production. They are valuable because present an opportunity to reuse waste materials and give some insight about the changes occurring in the process of aged asphalt rejuvenation and recycling. I found several shortcomings which are listed below. After their addressing I think that this paper could be published in Polymers journal. The paper, however, needs a major revision.
1. Page 2, line 45, the sentence: “Generally, 70–80% of REO can be effectively rejuvenated by the aforementioned processes.” Is difficult to understand for lack of explanation of the abbreviation REO.
2. Page 2, line 57, the expression: “…making its road performance more excellent” needs some recision, something like that: improving its road performance.”
3. SBS abbreviation is needed to explain in the paper, although it should be clear for the experts in the field of modified bitumen what it means.
4. Table 2.2. Left hand centering will remove the use of hyphen in the first left hand column.
5. The methods used to measure properties reported in Tables 1, and 2 is good to report in the tables.
6. It seems that the stabilizer used is elemental sulphur. Why don’t you write that the stabilizer used is elemental sulphur with 99.9% of sulphur.
7. Make a left hand centering in the left hand column of Table 2.3. Show the method used to measure properties reported in Table 2.3.
8. No data is reported about the viscosity of the raw asphalt in Table 2.1. to compare it with the viscosity of the aged asphalt shown in table 2.3. In this way the difference in viscosity a result from ageing process will be seen.
9. Page 4, line 112, “…add 6% SBS stabilizer…” The stabilizer is Sulphur or SBS. SBS is a component instead of stabilizer, after explaining in Section 2.1.4 Stabilizer, it is seen that the asphalt stabilizer used has an active ingredient of sulfur with a content of 99.9% or more. Thus, the stabilizer seems to be elemental Sulphur. Some confusion is obtained. More clarity is needed to add in the revised manuscript.
10. Explanation of meaning of the abbreviation AASHTO is missing in the manuscript.
11. Page 4, line 137, the sentence: “Among the S and m results…” is not clear for lack of explanation what “S” and “m” mean.
12. Page 5, line 162, the clause “…then the score value 162 Fi=Σ???×???? ????? ?????4 ?=1.” Has no predicate. There is no action.
13. The title of Figure 3. 1. “MSCR test results at 0.1kPa stress level” does not explain the titles of subfigures (a), and (b).
14. The same is valid for Figure 3.2.
15. Page 7, lines 193, 194, the expression: “REOB rejuvenated asphalt can effectively improve its stability performance under external” is not clear. This sentence does not end with a dot.
16. Fig.1 is advised to increase the size of letters and numbers to be visible. In the original manuscript they are too tiny. Interestingly Fig. 1 appears after Fig. 3.5. Probably it must be Fig. 3.6.
17. Typically the group composition of residual oils is expressed as saturates, aromatics, resins or polar aromatics, and asphaltenes. Here the authors use another designation of the four fraction SARA method, why?
18. Page 10, line 283, the sentence: “…which means that the saturated fraction and aromatic fraction of light oil fraction…” is not very clear because under light oil is typically considered the oils boiling below 360°C. Here, obviously it is not the case and a better and clearer expression is needed.
19. Page 10. The difference in SARA composition between the base asphalt and the aged asphalt is almost the same as that between the rejuvenated asphalt and the aged asphalt meaning that the process of rejuvenation restore the SARA composition of the aged asphalt. This needs to be highlighted in this section of the manuscript proving that the rejuvenation process restore the SARA compostion of the asphalt that is altered in the process of ageing.
20. Page 10, line 192, the sentence: “Base on the component blending theory…” needs a revision. It should be like this: “Based on the blending component theory…”
21. Fig.3.7, and 3.8. are too small. Increase of scale is recommended to become visible.

Author Response
Dear Professor,
I am very pleased to receive the letter from the reviewers' review of the manuscript " Preparation and macro-micro properties of SBS/REOB Modi-fied- Rejuvenated Asphalt" (ID: polymers-2035518). Thank you for your comments on our paper. These review comments are of high value to the revision and improvement of our manuscript, and also have important guiding significance for our research work. After carefully studying these comments and your advice, we have supplemented our analyses, and revised the manuscript content. Our responses to the comments are provided below.
Please refer to the attachment for details of the revised version
Point 1: Page 2, line 45, the sentence: “Generally, 70–80% of REO can be effectively rejuvenated by the aforementioned processes.” Is difficult to understand for lack of explanation of the abbreviation REO.
Response 1: Thanks to the reviewers for their suggestions. The REO in this sentence is recycled engine oil, which has been modified in page 2, line 54.
Point 2: Page 2, line 57, the expression: “…making its road performance more excellent” needs some recision, something like that: improving its road performance.”
Response 2: Thanks to the reviewers for pointing out the problem. We have modified this statement on page 2, line 68.
Point 3: SBS abbreviation is needed to explain in the paper, although it should be clear for the experts in the field of modified bitumen what it means.
Response 3: Thanks to the reviewer for the correction. SBS means Styrenic Block Copolymers, already changed on page 1, line 12.
Point 4: Table 2.2. Left hand centering will remove the use of hyphen in the first left hand column.
Response 4: Thank you for your comments, have been modified according to the comments.
Point 5: The methods used to measure properties reported in Tables 1, and 2 is good to report in the tables.
Response 5: Thanks to reviewers for their suggestions. The methods used to measure Tables 2.1 and 2.2 were chosen from the specifications, which are shown in the text at page 2, line 84.
Point 6: It seems that the stabilizer used is elemental sulphur. Why don’t you write that the stabilizer used is elemental sulphur with 99.9% of sulphur.
Response 6: Thanks to reviewers for pointing out the problem. Modified at A based on comments page 3, line 100.
Point 7: Make a left hand centering in the left hand column of Table 2.3. Show the method used to measure properties reported in Table 2.3.
Response 7: Thank you for your comments, have been modified according to the comments.
Point 8: No data is reported about the viscosity of the raw asphalt in Table 2.1. to compare it with the viscosity of the aged asphalt shown in table 2.3. In this way the difference in viscosity a result from ageing process will be seen.
Response 8: Thanks to reviewers for their suggestions. The relevant data have been added in Table 2.1.
Point 9: Page 4, line 112, “…add 6% SBS stabilizer…” The stabilizer is Sulphur or SBS. SBS is a component instead of stabilizer, after explaining in Section 2.1.4 Stabilizer, it is seen that the asphalt stabilizer used has an active ingredient of sulfur with a content of 99.9% or more. Thus, the stabilizer seems to be elemental Sulphur. Some confusion is obtained. More clarity is needed to add in the revised manuscript.
Response 9: Thanks to reviewers for pointing out the problem. In this paper, the amount of stabilizer added should be 6% of the amount of SBS added, This question has been changed in the text at page 4 line 126-127.
Point 10: Explanation of meaning of the abbreviation AASHTO is missing in the manuscript.
Response 10: Thanks to reviewers for pointing out the problem. AASHTO full name American Association of State Highway and Transportation Officials, has been revised in the text at page 5 line 141.
Point 11: Page 4, line 137, the sentence: “Among the S and m results…” is not clear for lack of explanation what “S” and “m” mean.
Response 11: Thanks to the reviewer for the correction. The meaning of “S” and “m” is explained in the text at page 5 line 152.
Point 12: Page 5, line 162, the clause “…then the score value 162 Fi=Σ???×???? ????? ?????4 ?=1.” Has no predicate. There is no action.
Response 12: Thanks to reviewers for pointing out the problem. This issue you raised has been revised at page 5 line 179.
Point 13: The title of Figure 3. 1. “MSCR test results at 0.1kPa stress level” does not explain the titles of subfigures (a), and (b).
Response 13: Thanks to the reviewers for their suggestions. This issue has been changed in the title of Figure 3.1
Point 14: The same is valid for Figure 3.2.
Response 14: Thanks to the reviewers for pointing out the problem. This issue has been changed in the title of Figure 3.2
Point 15: Page 7, lines 193, 194, the expression: “REOB rejuvenated asphalt can effectively improve its stability performance under external” is not clear. This sentence does not end with a dot.
Response 15: Thanks to the reviewer for the correction. The sentence has been revised at page 7 line 217-219.
Point 16: Fig.1 is advised to increase the size of letters and numbers to be visible. In the original manuscript they are too tiny. Interestingly Fig. 1 appears after Fig. 3.5. Probably it must be Fig. 3.6.
Response 16: Thank you for your comments, have enlarged the size of the letters and numbers in the figure. and modified the figure number to Figure 3.6
Point 17: Typically the group composition of residual oils is expressed as saturates, aromatics, resins or polar aromatics, and asphaltenes. Here the authors use another designation of the four fraction SARA method, why?
Response 17: Thanks to reviewers for their suggestions. The original statement used by the author was inaccurate and has been corrected in the text at page 10 line 314, etc.
Point 18: Page 10, line 283, the sentence: “…which means that the saturated fraction and aromatic fraction of light oil fraction…” is not very clear because under light oil is typically considered the oils boiling below 360°C. Here, obviously it is not the case and a better and clearer expression is needed.
Response 18: Thanks to reviewers for pointing out the problem. Here should be light component instead of light oil fraction, has been modified in the text of page 10 line 316.
Point 19: Page 10. The difference in SARA composition between the base asphalt and the aged asphalt is almost the same as that between the rejuvenated asphalt and the aged asphalt meaning that the process of rejuvenation restore the SARA composition of the aged asphalt. This needs to be highlighted in this section of the manuscript proving that the rejuvenation process restore the SARA compostion of the asphalt that is altered in the process of ageing.
Response 19: Thank you for your comments, This part has been emphasized in the text at page 10 line 324.
Point 20: Page 10, line 192, the sentence: “Base on the component blending theory…” needs a revision. It should be like this: “Based on the blending component theory…”
Response 20: Thanks to reviewers for their suggestions. This part has been revised in the text at page 10 line 327.
Point 21: Fig.3.7, and 3.8. are too small. Increase of scale is recommended to become visible.
Response 21: Thanks to reviewers for pointing out the problem. have enlarged the size of the letters and numbers in the fig.3.7, and 3.8.

Round 2
Reviewer 1 Report
The authors have addressed my comments. I have no further comments on the paper.
Reviewer 2 Report
The revised manuscript is acceptable.
Reviewer 3 Report
The authors have adequately addressed reviewer’s comments and properly revised the manuscript. Therefore, it is acceptable for publication in the journal Polymers.